# Learning Rotation-Invariant Representation using Rotation-Equivariant CNNs

## Abstract

Conventional self-supervised learning (SSL) methods, such as SimCLR and Sim-Siam, have demonstrated significant effectiveness. However, their feature representation is not robust to image rotations, as rotational augmentation may negatively impact the framework. In this paper, we address this limitation by applying SSL to group-equivariant CNNs, specifically rotation-equivariant CNNs, to develop robust features. To learn expressive, rotation-invariant features, we introduce our training method, Guiding Invariance with Equivariance (GIE), which simultaneously trains both invariant features and the equivariance score for images. The equivariance score guides the rotation-equivariant features through an attention-weighted sum mechanism, enabling the development of rotation-invariant features. Through experiments, we demonstrate that our GIE method not only extracts high-performing features under four discrete rotations but also achieves robustness to random-degree rotations through rotation augmentation training. These results highlight the effectiveness of our method in achieving robust rotation-invariance.

## 1 Introduction

How do humans recognize rotated images? Although it may seem simple and straightforward for humans to recognize rotated images, Figure 1 shows that this is not always the case. When attempting to read rotated text, we don't read it directly but rather follow a sequential process. In this process, we first try to understand how the image is rotated, then mentally "rotate" it back to its original position before reading. Similarly, when identifying objects in a rotated image, our brain doesn't immediately recognize the object. Instead, it analyzes the image, determining the angle at which the recognizable "object" emerges, and then mentally rotates it back to its correct orientation before accurately identifying the object. In some cases, such as with the number 9, it can be difficult to determine how many degrees the image has been rotated, making it challenging to accurately recognize the number. The difficulty in analyzing such samples is a natural phenomenon and supports the claim that humans perform a sequential process when analyzing rotated images.

However, in deep learning, analyzing and rotating an image to its original position before extracting features requires using the model twice, which is resource-intensive. With this motivation, we propose the Guiding Invariance with Equivariance (GIE) method, which applies the sequential process at the feature level rather than the image level (see Figure 1). We used a rotation-equivariant model as the feature extractor, ensuring that the output features behave equivariantly with respect to the input image's rotation. From these features, we obtained an equivariance score that indicates the degree of image rotation, allowing us to apply the sequential process at the feature level, similar to how humans analyze images. Through this process, we can naturally extract rotation-invariant features guided by the equivariance scores.

We conducted experiments using a self-supervised learning (SSL) approach to train the rotation-invariant features extracted by the GIE method and evaluated the model across various experimental datasets. We tested two SSL methods, SimCLR (Chen et al., 2020) and SimSiam (Chen & He, 2021), using CIFAR10 (Krizhevsky et al., 2009), STL10 (Coates et al., 2011), and ImageNet100 (Tian et al., 2020) as datasets. The experimental results with the two SSL methods and three datasets showed that, in all cases, the rotation-invariant features extracted by the GIE method achieved higher linear classification accuracy for 0, 90, 180, and 270-degree rotations compared to other baseline feature

Figure 1: Human recognition of rotated images and the concept behind the GIE method. Humans do not directly recognize a rotated image but instead process it sequentially. Motivated by this, we propose the GIE method, which employs a sequential process at the feature level.

extractors, such as basic CNN and E(2)-CNN (Weiler & Cesa, 2019) models. Furthermore, we added rotation augmentation to the input image transform to extract rotation-invariant features for random angles between 0 and 360 degrees. Through these experiments, we were able to observe that our GIE method achieved more stable and higher linear accuracy across all degrees compared to other baselines.

Additionally, we analyzed the equivariance score across various datasets to understand its significance. Through extensive experiments, we found that the equivariance score effectively guides the features to a recognizable relative orientation, aligning with our intended concept. Furthermore, we extended our experiments to include different rotation group orders and conducted experiments on dense prediction tasks.

To summarize:

- We introduce the Guiding Invariance with Equivariance (GIE) method as a novel approach for learning superior rotation-invariant representations using rotation-equivariant CNNs.

- Using the GIE method, we generated rotation-invariant features and demonstrated robust performance against rotations by applying them to various self-supervised learning methods and image datasets.

- We analyzed the significance of the equivariance score through various experiments.

- We proposed several extensions, including different group orders and dense prediction.

## 2 RELATED WORK

SSL techniques leverage diverse augmentations and learn their underlying similarities to efficiently extract representations from unlabeled data. Among SSL techniques, there are pretext-based models like RotNet (Gidaris et al., 2018), which predicts rotated images, and contrastive learning-based models such as SimCLR (Chen et al., 2020) and SimSiam (Chen & He, 2021). Additionally, E-SSL (Dangovski et al., 2022) combines a separate module that predicts rotated images, like RotNet, with contrastive learning-based models to extract equivariant features. Using these models as base-

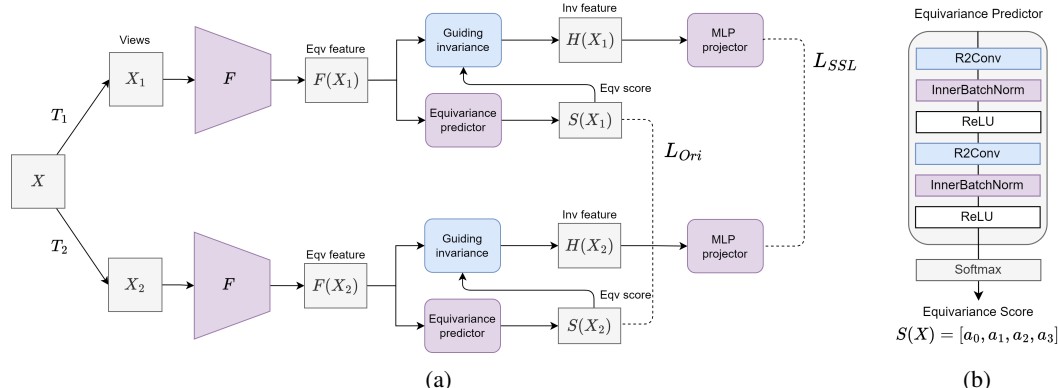

Figure 2: (a) Architecture of the GIE method and (b) Architecture of the equivariance predictor. (a) We utilized the rotation-invariant feature $H(X)$, created through guiding invariance, as a feature encoder for contrastive learning. (b) Specifically, 'R2Conv,' 'InnerBatchNorm,' and 'ReLU' correspond to the $1 \times 1$ group-equivariant convolution, batch normalization, and ReLU layer, respectively, from the e2cnn library (Cesa et al., 2021), which preserves the equivariance of the input feature.

lines, we propose a rotation-invariant representation by integrating an equivariance predictor into the contrastive learning-based models of SimCLR and SimSiam.

Group-equivariant convolutional neural network (GCNNs) first appeared in the work by Cohen & Welling (2016) and demonstrated good performance on rotated MNIST (Ghifary et al., 2015) due to their property of being equivariant to image transformation groups. Subsequent research (Weiler & Cesa, 2019; Cohen & Welling, 2017; Hoogeboom et al., 2018; Cohen et al., 2019) has further explored the properties of GCNNs. Studies using GCNNs to address problems related to rotation transforms have been conducted in numerous paper (Worrall et al., 2017; Weiler et al., 2018b; Bekkers et al., 2018; Marcos et al., 2017; Weiler et al., 2018a), achieving effective performance across various fields. We design a group convolutional neural network equivariant to the $p4$-group, similar in structure to ResNet (He et al., 2016), using e2cnn library (Cesa et al., 2021).

In the field of representation learning, various techniques have been suggested for equivariant representations. Many methods (Garrido et al., 2023; Dangovski et al., 2022; Feng et al., 2019; Gidaris et al., 2018; Lee et al., 2021; Bai et al., 2023; Xu & Triesch, 2023; Devillers & Lefort, 2022) incorporate a predictor that matches the encoded augmented data to extract equivariant properties with respect to the given transform(e.g., determining the degree of rotation). Another approach (Lee et al., 2023) involves using the ReResNet (Han et al., 2021) encoder, which can extract equivariant features without performing rotation transforms. In our work, similar to Han et al. (2021), we used an E(2)-CNN backbone network following the ResNet architecture.

## 3 LEARNING ROTATION-EQUIVARIANT CNNS

### 3.1 OVERVIEW

In Figure 2a, we have illustrated our GIE method. To obtain rotation-invariant features, we employed a rotation-equivariant backbone network $F$ to extract the rotation-equivariant feature $F(X)$. Unlike previous SSL approaches, we designed an equivariance predictor module using a $1 \times 1$ group convolution layer from the e2cnn library (Cesa et al., 2021) to maintain equivariance of the equivariance score, where the score refers to the output of the equivariance predictor. We assigned an orientation alignment loss to facilitate the learning of the equivariance score. Using the learned equivariance score and the rotation-equivariant features extracted from the backbone, we conducted a guiding invariance process to create rotation-invariant features. For the training loss functions, we combine the conventional SSL loss with the orientation alignment loss. Also, the background information related to group-equivariant CNNs mentioned here is presented in Appendix A.1

## 3.2 GROUP-EQUIVARIANT CNNs

Group-equivariant CNNs (GCNNs) maintain equivariance with a predefined image transformation group $G$, making them effective for extracting equivariant features. We used a cyclic group of order 4 (i.e., $p4$-group), corresponding to 90-degree rotations, as our group $G$. Our backbone model $F$ is based on the ResNet architecture, with the layers replaced by equivariant layers provided by the e2cnn library.

The output feature vector exhibits equivariance, where the rotation of the input image corresponds to a permutation in the group dimension of the feature vector. Formally, let $X$ represent the input image, and $F(X) \in \mathbb{R}^{|G| \times K}$ be the equivariant feature passed through the backbone $F$. We define $G$ is $p4$-group, hence $|G| = 4$. Then, $F(X)$ can be expressed as follows:

$$F(X) := [f_0, f_1, f_2, f_3], \qquad f_i \in \mathbb{R}^K \tag{1}$$

Since $F(X)$ exhibits rotation-equivariance, for $rX, r^2X, r^3X$, which represent the input $X$ rotated by $90°, 180°, 270°$, respectively, we have the following relationships:

$$F(rX) = F(r^{-3}X) = [f_3, f_0, f_1, f_2] \tag{2}$$

$$F(r^2X) = F(r^{-2}X) = [f_2, f_3, f_0, f_1] \tag{3}$$

$$F(r^3X) = F(r^{-1}X) = [f_1, f_2, f_3, f_0] \tag{4}$$

Due to the characteristics of the equivariant backbone model, we can replace image rotation with a feature-level permutation.

Edixhoven et al. (2023) analyze the exact equivariance of GCNNs. As mentioned, since typical GC-NNs do not achieve exact equivariance, we adjusted the input image size following the mathematical conditions from Edixhoven et al. (2023) to ensure exact equivariance in the final output feature vector. By setting this, we ensured that the final output feature vector maintained exact equivariance. The details are as follows.

**Exact equivariance in GCNN on $p4$-group (Edixhoven et al., 2023)** A GCNN is exactly equivariant to rotations of multiples of 90-degree if the following equation holds for all layers in the network:

$$(i - k) \mod s = 0. \tag{5}$$

where $i$ is the rectangular image size, $k$ is the kernel size and $s$ is the stride. Based on the previous equation, we reshaped the images in each dataset to fit the model architecture in the experiments.

## 3.3 EQUIVARIANCE PREDICTOR

After the feature vector is extracted, it is fed into an equivariance predictor (see Figure 2b). The predictor consists of a $1 \times 1$ group-equivariant convolution, batch normalization, and a ReLU layer, all designed to preserve group-equivariance using the e2cnn library. This design enables the equivariance predictor to analyze the input feature vector effectively while maintaining the rotation-equivariance of the output. We set the final output dimension to a 1-regular representation (4 dimensions), which enabled the generation of an equivariant 4-dimensional score, referred to as the equivariance score.

The distinctive characteristic of the equivariance score is its rotation-equivariance; if the input image is rotated by 90 degrees, the original equivariance score shifts laterally by one position. Consequently, once the equivariance score $S(X)$ for an input image $X$ is determined, the scores for the rotated images $S(rX), S(r^2X)$, and $S(r^3X)$ are automatically defined due to their rotational equivariance. In other words, if we define $S(X) := [a_0, a_1, a_2, a_3]$, where $a_i \in \mathbb{R}$ are scalar values, then $S(rX), S(r^2X)$, and $S(r^3X)$ are determined as $[a_3, a_0, a_1, a_2], [a_2, a_3, a_0, a_1]$, and $[a_1, a_2, a_3, a_0]$, respectively. Thus, a 4-dimensional vector effectively captures the properties of these four rotated image states (see Figure 3).

## 3.4 GUIDING INVARIANCE

Using the equivariance score $S(X)$, we create the rotation-invariant feature $H(X)$ by guiding the original feature $F(X)$. We refer to this process as *guiding invariance*. For the guiding invariance component, we selected *group attentioning* operation to conduct our experiments.

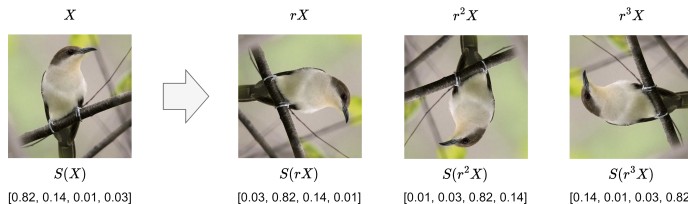

Figure 3: An example of an equivariance score. If the input image is rotated, the equivariance score cyclically shifts by one position.

**Group attentioning** Let $X$ as the input image, $F(X) \in \mathbb{R}^{|G| \times K}$ as the equivariant feature passed through the backbone, and $S(X) \in \mathbb{R}^{|G|}$ as the equivariance score. We set $G$ as a $p4$-group, hence $|G| = 4$. Then $F(X), S(X)$ can be expressed as follows:

$$F(X) := [f_0, f_1, f_2, f_3], \quad S(X) := [a_0, a_1, a_2, a_3], \qquad f_i \in \mathbb{R}^K, a_i \in \mathbb{R} \tag{6}$$

We define our rotation invariant feature $H(X)$ as follows:

$$\begin{aligned} H(X) &:= S(X) \cdot [F(X), F(r^{-1}X), F(r^{-2}X), F(r^{-3}X)] \\ &= a_0 F(X) + a_1 F(r^{-1}X) + a_2 F(r^{-2}X) + a_3 F(r^{-3}X) \end{aligned} \tag{7}$$

Then, the feature $H(X)$ is rotation-invariant, as demonstrated by:

$$\begin{aligned} H(rX) &= S(rX) \cdot [F(rX), F(X), F(r^{-1}X), F(r^{-2}X)] \\ &= a_3 F(rX) + a_0 F(X) + a_1 F(r^{-1}X) + a_2 F(r^{-2}X) \\ &= a_3 F(r^{-3}X) + a_0 F(X) + a_1 F(r^{-1}X) + a_2 F(r^{-2}X) \\ &= H(X) \end{aligned} \tag{8}$$

From the definition of $H(X)$, it is evident that $S(X)$ acts like attention weights on the encoder features of each rotated image. Consequently, we refer to this operation as *group attentioning*. Consistent with the concept of attention as described in Woo et al. (2018), we applied a softmax function across the group dimension to the output of the equivariance predictor. The related pseudo code is provided in Algorithm 1 of Appendix A.2.

### 3.5 LOSS FUNCTION

To simultaneously train the rotation-invariant feature and the equivariance score, we combined the loss used in traditional self-supervised learning with another loss designed for equivariance score training. We introduce the orientation alignment loss that we used for training.

**Orientation alignment loss** The orientation alignment loss, as used in Lee et al. (2023), ensures that the equivariance scores of different image views match. We have simplified the orientation alignment loss since we do not need to align the orientations of the images.

Let $X$ be the input image, and define $X_1$, $X_2$ as the outputs of different transformations $T_1$, $T_2$ applied to $X$ (i.e., $X_i = T_i(X), i = 1, 2$. see Figure 2a). We define our orientation alignment loss as follows:

$$L_{Ori}(X_1, X_2) = -\sum_{i=1}^{4} S(X_1)_i \log(S(X_2)_i) \tag{9}$$

This is essentially the cross-entropy loss between $S(X_1)$ and $S(X_2)$. Unlike the method used in the original Lee et al. (2023), since the dominant orientation in our training dataset is aligned to 0 degrees, we used the cross-entropy loss without any additional shift operations.

**Total loss** We integrate the conventional self-supervised learning loss (SSL loss) with our equivariance loss to form the loss function. For SimCLR, the SSL loss is infoNCE (Oord et al., 2018),

and for SimSiam, it is negative cosine similarity loss. Let $L_{SSL}$ be the SSL loss and $L_{Ori}$ be the orientation alignment loss, then the total loss $L$ is defined as follows:

$$L := L_{SSL} + \beta \cdot L_{Ori} \tag{10}$$

Here, $\beta$ is a scalar weight assigned to the $L_{Ori}$. The details regarding the choice of $\beta$ are covered in Appendix B.4.

## 4 EXPERIMENTS

### 4.1 SETUPS

We applied our GIE method to both SimCLR and SimSiam to demonstrate its robustness across various SSL frameworks, without dependency on a specific method. To validate the effectiveness of GIE, we compared the results to baseline SimCLR and SimSiam methods, where each SSL method was trained with a ResNet backbone. Additionally, we compared our approach to SimCLR and SimSiam experiments using an E(2)-CNN backbone, structured similarly to ResNet but without the application of GIE. Since E(2)-CNN backbones generally consume more GPU memory during training compared to conventional CNNs, we adjusted the model size of the E(2)-CNN backbone (by modifying the number of channels, depth, etc.) to ensure comparable or reduced GPU memory consumption relative to a standard ResNet backbone. Furthermore, we included RotNet and E-SSL, which learn features by predicting image rotations, as additional baseline comparisons. These methods were chosen due to their alignment with our approach, as both predict image rotations similar to our goal of training the equivariance score, which effectively learns features from rotated images.

Our experimental evaluation was conducted on datasets including CIFAR10, STL10, and ImageNet100. For further details and training settings, please refer to Appendix B.1

### 4.2 ROTATION INVARIANCE ACROSS FOUR DISCRETE 90-DEGREE ORIENTATIONS

For the CIFAR10 dataset, we conducted the following experiments. Contrastive learning methods such as SimCLR and SimSiam, SSL methods trained with rotation prediction loss such as RotNet (Gidaris et al., 2018) and E-SSL (Dangovski et al., 2022), contrastive learning methods with the backbone replaced by the E(2)-CNN architecture, and our proposed GIE method. For SimCLR and SimSiam, we used ResNet18 as the backbone. Unlike the standard augmentation transforms in SimCLR and SimSiam, we added 90-degree four-direction rotation augmentation to create new experiments (SimCLR(R) and SimSiam(R)). In the case of RotNet, we conducted experiments using two backbones: Network in Network (NIN, Lin (2013)) and ResNet18. For E-SSL, we experimented with both SimCLR and SimSiam. For SimCLR and SimSiam with the E(2)-CNN backbone, we experimented with two setups: one with group pooling from the e2cnn library added to the final layer and one without group pooling. This was done to compare the traditional group pooling method for extracting rotation-invariant features with our proposed GIE method. The E(2)-CNN backbone followed the depth and layer structure of ResNet18, with the number of channels adjusted to ensure no significant difference in training GPU memory consumption compared to ResNet18 (see Appendix B.5). As shown in Table 1, our E(2)-CNN based experiments consumed similar GPU memory compared to other networks, while recording a lower number of encoder parameters. In all experiments, the backbone architecture was adjusted to match the image size of CIFAR10 by modifying the stride.

After pretraining, we froze the pretrained backbone and attached a linear classifier to measure linear classification accuracy. Additionally, to assess rotation invariance across four directions, we evaluated the linear classification accuracy on both the Non-Rotated (NR) dataset and the Rotated (R) dataset, which included images rotated in four directions. As shown in Table 1, our GIE method achieved the highest linear evaluation performance on the R dataset for both SimCLR and SimSiam, while also recording comparable performance on the NR dataset. Furthermore, even in the experiments using the E(2)-CNN backbone, the GIE method outperformed the other two cases where GIE was not applied. These results demonstrate the clear advantages of GIE as a training method.

For STL10 training, we used a setting similar to that of CIFAR10, with slight modifications to accommodate the STL10 image size. For the baseline models SimCLR and SimSiam, we used two

Table 1: Results on CIFAR10, STL10, and ImageNet100. We conducted training using various SSL methods and different backbones. 'EqvPred' refers to our equivariance predictor. Since we use the $H(X)$ feature, the equivariance predictor is conceptually included in the backbone encoder.

| Dataset | SSL | Method | Backbone | GPU Memory (GB) | Encoder Params (M) | NR | R |
|---|---|---|---|---|---|---|---|
| CIFAR10 | RotNet | RotNet | NIN | 6.1 | 1.41 | 89.69 | 86.70 |
| | | RotNet | ResNet18 | 4.7 | 11.17 | 87.75 | 86.86 |
| | SimCLR | SimCLR | ResNet18 | 6.7 | 11.17 | 91.47 | 72.20 |
| | | SimCLR(R) | ResNet18 | 6.7 | 11.17 | 86.52 | 86.43 |
| | | E-SSL | ResNet18 | 8.8 | 11.17 | **93.57** | 81.58 |
| | | SimCLR | E(2)-CNN | 8.8 | 2.93 | 91.63 | 87.89 |
| | | SimCLR | E(2)-CNN + Group Pooling | 8.8 | 2.93 | 90.25 | 90.27 |
| | | SimCLR + GIE (ours) | E(2)-CNN + EqvPred | 8.6 | 3.26 | 91.72 | **92.01** |
| | SimSiam | SimSiam | ResNet18 | 6.7 | 11.17 | 91.14 | 71.80 |
| | | SimSiam(R) | ResNet18 | 6.7 | 11.17 | 86.14 | 86.29 |
| | | E-SSL | ResNet18 | 8.8 | 11.17 | **93.76** | 83.29 |
| | | SimSiam | E(2)-CNN | 8.8 | 2.93 | 90.82 | 86.75 |
| | | SimSiam | E(2)-CNN + Group Pooling | 8.8 | 2.93 | 90.60 | 90.46 |
| | | SimSiam + GIE (ours) | E(2)-CNN + EqvPred | 8.6 | 3.26 | 91.05 | **91.08** |
| STL10 | RotNet | RotNet | ResNet18 | 4.5 | 11.18 | 76.57 | 76.39 |
| | SimCLR | SimCLR | ResNet18 | 5.5 | 11.18 | 83.34 | 69.90 |
| | | SimCLR(R) | ResNet18 | 5.5 | 11.18 | 77.48 | 75.47 |
| | | SimCLR | ResNet50 | 12.6 | 23.51 | 87.56 | 75.84 |
| | | SimCLR(R) | ResNet50 | 12.6 | 23.51 | 83.13 | 82.90 |
| | | E-SSL | ResNet50 | 19.0 | 23.51 | **87.68** | 77.43 |
| | | SimCLR | E(2)-CNN | 9.6 | 11.14 | 86.50 | 82.11 |
| | | SimCLR | E(2)-CNN + Group Pooling | 9.6 | 11.14 | 85.05 | 84.08 |
| | | SimCLR + GIE (ours) | E(2)-CNN + EqvPred | 9.1 | 12.83 | 86.48 | **86.44** |
| | SimSiam | SimSiam | ResNet18 | 5.5 | 11.18 | 84.46 | 71.52 |
| | | SimSiam(R) | ResNet18 | 5.5 | 11.18 | 73.05 | 72.77 |
| | | SimSiam | ResNet50 | 12.6 | 23.51 | 84.91 | 72.18 |
| | | SimSiam(R) | ResNet50 | 12.6 | 23.51 | 74.56 | 74.30 |
| | | E-SSL | ResNet50 | 17.9 | 23.51 | 85.99 | 75.85 |
| | | SimSiam | E(2)-CNN | 9.6 | 11.14 | 86.01 | 84.15 |
| | | SimSiam | E(2)-CNN + Group Pooling | 9.6 | 11.14 | 82.68 | 84.19 |
| | | SimSiam + GIE (ours) | E(2)-CNN + EqvPred | 9.1 | 12.83 | **87.41** | **88.31** |
| ImageNet100 | SimCLR | SimCLR | ResNet50 | 28.20 | 23.51 | **76.06** | 66.37 |
| | | SimCLR(R) | ResNet50 | 28.20 | 23.51 | 72.24 | 71.84 |
| | | SimCLR | E(2)-CNN | 20.05 | 11.14 | 72.42 | 68.43 |
| | | SimCLR | E(2)-CNN + Group Pooling | 20.05 | 11.14 | 70.60 | 70.20 |
| | | SimCLR + GIE (ours) | E(2)-CNN + EqvPred | 20.24 | 12.83 | 73.34 | **72.75** |
| | SimSiam | SimSiam | ResNet50 | 28.43 | 23.51 | 73.42 | 61.85 |
| | | SimSiam(R) | ResNet50 | 28.43 | 23.51 | 68.30 | 71.19 |
| | | SimSiam | E(2)-CNN | 20.56 | 11.14 | 75.10 | 73.09 |
| | | SimSiam | E(2)-CNN + Group Pooling | 19.65 | 11.14 | 71.82 | 73.26 |
| | | SimSiam + GIE (ours) | E(2)-CNN + EqvPred | 18.80 | 12.83 | **75.62** | **76.54** |

backbone models: ResNet18 and ResNet50. In the case of the E(2)-CNN backbone, we followed the ResNet18 structure but increased the number of channels, making the model larger than the one used for CIFAR10 while keeping the GPU memory cost below that of the ResNet50 model. Similar to the CIFAR10 results, as shown in Table 1, our GIE method achieved the highest performance on the R dataset, while also showing comparable results on the NR dataset.

Based on the results from CIFAR10 and STL10, we extended our experiments to the large-scale image dataset, ImageNet100. Since the performance of RotNet and E-SSL on CIFAR10 and STL10 was lower than that of the baseline experiments SimCLR(R) and SimSiam(R), we excluded them from the baseline comparisons. The experimental results showed that our GIE method achieved the highest performance on the 4-direction rotated dataset and recorded comparable results on the non-rotated dataset, as shown in Table 1.

### 4.3 ROTATION INVARIANCE UNDER ARBITRARY-DEGREE ROTATIONS

We conducted experiments to evaluate rotation-invariance for random degrees. One issue that arises with square images is the distortion of edges when rotated at non-90-degree intervals. To address this, we applied a circular crop to the images during transformation, ensuring uniform information across all rotations. Additionally, during pretraining, we incorporated random rotation augmentation to allow the model to learn features across all angles. For methods using the E(2)-CNN backbone, we applied rotation augmentation within the range of -45 to 45 degrees, as these methods exhibit periodicity at 90-degree intervals. Similarly, for RotNet and E-SSL, we applied rotation augmentation within the -45 to 45 degree range, aligning with their concept of rotation prediction. In contrast,

Table 2: Arbitrary-degree rotations results on CIFAR10, STL10, and ImageNet100. We trained using circular crop transformations across various experimental settings, then measured the linear classification accuracy on datasets rotated in 5-degree increments. The reported values represent the mean and standard deviation of the linear classification accuracy across different angles.

| Dataset | SSL | Method | Backbone | Rotation Augmentation Degree | (0,5,10,…,355) R |
|---|---|---|---|---|---|
| CIFAR10 | RotNet | RotNet | NIN | $[-45°, 45°]$ | $81.609 \pm 0.344$ |
| | | RotNet | ResNet18 | $[-45°, 45°]$ | $82.812 \pm 0.516$ |
| | SimCLR | SimCLR(R) | ResNet18 | $[0°, 360°]$ | $82.498 \pm 0.245$ |
| | | E-SSL | ResNet18 | $[-45°, 45°]$ | $79.974 \pm 4.972$ |
| | | SimCLR | E(2)-CNN | $[-45°, 45°]$ | $85.503 \pm 0.270$ |
| | | SimCLR | E(2)-CNN + Group Pooling | $[-45°, 45°]$ | $86.088 \pm 0.255$ |
| | | SimCLR + GIE (ours) | E(2)-CNN + EqvPred | $[-45°, 45°]$ | $\mathbf{86.750 \pm 0.177}$ |
| | SimSiam | SimSiam(R) | ResNet18 | $[0°, 360°]$ | $82.181 \pm 0.234$ |
| | | E-SSL | ResNet18 | $[-45°, 45°]$ | $78.007 \pm 5.194$ |
| | | SimSiam | E(2)-CNN | $[-45°, 45°]$ | $83.843 \pm 0.244$ |
| | | SimSiam | E(2)-CNN + Group Pooling | $[-45°, 45°]$ | $86.309 \pm 0.162$ |
| | | SimSiam + GIE (ours) | E(2)-CNN + EqvPred | $[-45°, 45°]$ | $\mathbf{88.917 \pm 0.300}$ |
| STL10 | RotNet | RotNet | ResNet18 | $[-45°, 45°]$ | $68.114 \pm 0.685$ |
| | SimCLR | SimCLR(R) | ResNet18 | $[0°, 360°]$ | $74.171 \pm 0.232$ |
| | | SimCLR(R) | ResNet50 | $[0°, 360°]$ | $80.492 \pm 0.202$ |
| | | E-SSL | ResNet50 | $[-45°, 45°]$ | $76.705 \pm 3.355$ |
| | | SimCLR | E(2)-CNN | $[-45°, 45°]$ | $80.863 \pm 0.189$ |
| | | SimCLR | E(2)-CNN + Group Pooling | $[-45°, 45°]$ | $81.379 \pm 0.168$ |
| | | SimCLR + GIE (ours) | E(2)-CNN + EqvPred | $[-45°, 45°]$ | $\mathbf{83.548 \pm 0.319}$ |
| | SimSiam | SimSiam(R) | ResNet18 | $[0°, 360°]$ | $69.658 \pm 0.134$ |
| | | SimSiam(R) | ResNet50 | $[0°, 360°]$ | $73.217 \pm 0.158$ |
| | | E-SSL | ResNet50 | $[-45°, 45°]$ | $67.397 \pm 3.768$ |
| | | SimSiam | E(2)-CNN | $[-45°, 45°]$ | $78.774 \pm 0.190$ |
| | | SimSiam | E(2)-CNN + Group Pooling | $[-45°, 45°]$ | $79.834 \pm 0.127$ |
| | | SimSiam + GIE (ours) | E(2)-CNN + EqvPred | $[-45°, 45°]$ | $\mathbf{82.160 \pm 0.254}$ |
| ImageNet100 | SimCLR | SimCLR(R) | ResNet50 | $[0°, 360°]$ | $70.60 \pm 0.27$ |
| | | SimCLR | E(2)-CNN | $[-45°, 45°]$ | $70.18 \pm 0.32$ |
| | | SimCLR | E(2)-CNN + Group Pooling | $[-45°, 45°]$ | $71.10 \pm 0.28$ |
| | | SimCLR + GIE (ours) | E(2)-CNN + EqvPred | $[-45°, 45°]$ | $\mathbf{73.15 \pm 0.66}$ |
| | SimSiam | SimSiam(R) | ResNet50 | $[0°, 360°]$ | $63.33 \pm 0.20$ |
| | | SimSiam | E(2)-CNN | $[-45°, 45°]$ | $69.16 \pm 0.27$ |
| | | SimSiam | E(2)-CNN + Group Pooling | $[-45°, 45°]$ | $70.08 \pm 0.26$ |
| | | SimSiam + GIE (ours) | E(2)-CNN + EqvPred | $[-45°, 45°]$ | $\mathbf{72.89 \pm 0.33}$ |

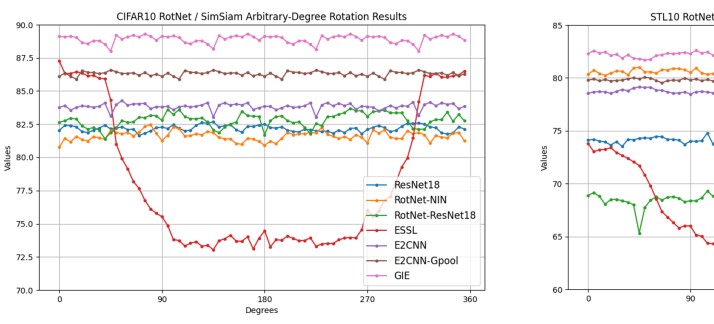

(a) RotNet/SimSiam results for CIFAR10 dataset   (b) RotNet/SimSiam results for STL10 dataset

Figure 4: SimSiam CIFAR10/STL10 results for arbitrary-degree rotations. For additional settings, please refer to Figure 11 for the corresponding graphs.

for the baseline SimCLR and SimSiam methods, we used random rotation augmentation from 0 to 360 degrees to ensure uniform learning across all angles. After pretraining, we attached a linear classifier and trained it on images rotated across all angles from 0 to 360 degrees.

To evaluate performance on fine-grained rotations, we rotated the images in 5-degree increments and measured the linear classification accuracy. The results in Table 2 indicate that, in both SimCLR and SimSiam settings, as well as across the CIFAR10, STL10, and ImageNet100 datasets, our GIE method achieved the highest mean accuracy with a low standard deviation, demonstrating its stability across all angles. As shown in Table 2 and Figure 4, the GIE method consistently outperformed other approaches across all rotation degrees.

Table 3: Dominance ratios across different datasets. We examined the distribution of equivariance scores for various datasets using the GIE model trained on ImageNet100. A value exceeding 0.97 in any dimension was designated as 'dominant.' The highest proportion for each dataset is highlighted in bold.

| Ratio | ImageNet100 | STL10 | Caltech256 | Stanford cars | FGVC-Aircraft | CUB-200-2011 | Oxford 102 Flowers | MTARSI |
|---|---|---|---|---|---|---|---|---|
| **Dimension 1 Dominant (> 0.97)** | 0.83% | 0.18% | 0.99% | 0.00% | 0.00% | 0.72% | 1.08% | 9.28% |
| **Dimension 2 Dominant (> 0.97)** | 1.49% | 0.14% | 1.55% | 0.00% | 0.01% | 0.20% | 0.72% | 5.68% |
| **Dimension 3 Dominant (> 0.97)** | 0.17% | 0.32% | 0.90% | 0.01% | 0.01% | 0.77% | 0.89% | 6.97% |
| **Dimension 4 Dominant (> 0.97)** | 89.01% | 91.34% | 62.65% | 99.51% | 98.92% | 81.36% | 28.40% | 8.11% |
| **Non-dominant (all dimensions ≤ 0.97)** | 8.49% | 8.02% | 33.92% | 0.48% | 1.05% | 16.95% | **68.91%** | **69.95%** |

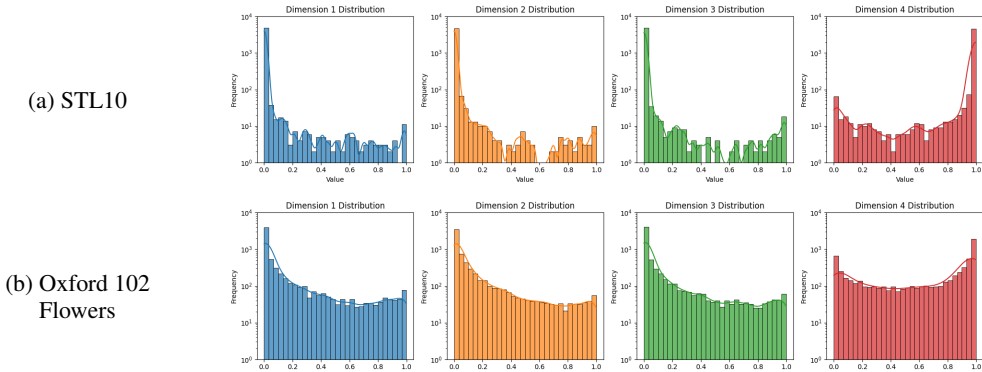

(a) STL10

(b) Oxford 102 Flowers

Figure 5: Equivariance score distribution histogram. For STL10, the majority of samples are concentrated in the 0.97-1.0 bin of dimension 4. In contrast, the Oxford 102 Flowers dataset shows a relatively uniform distribution. Histograms for other datasets can be found in Figure 12.

## 4.4 FURTHER STUDY

**Analysis of equivariance score** We examined the distribution of the equivariance score for the pretrained model using the GIE method on the ImageNet100 dataset (see Table 3 and Figure 5). Although we did not use any loss function that amplifies a particular dimension (e.g., rotation prediction loss), the equivariance score was dominated by specific dimensions, with a majority of the samples showing dominant values in these dimensions. We refer to these scores as *dominant scores*. This phenomenon aligns well with our motivation and intent for the equivariance score to reflect the relative orientation in which an image is most easily recognized.

Additionally, by analyzing the equivariance score, we gained insights into the overall characteristics of different datasets. As shown in Table 3, the STL10 and the ImageNet100 exhibited dominant score proportions of 91.34% and 89.01%, respectively, indicating a strong bias toward specific orientations. In contrast, the Oxford 102 Flowers (Nilsback & Zisserman, 2008) dataset, which contains more rotation-invariant images, showed a dominant score proportion of only 28.40%. These results demonstrate that the equivariance score effectively captures both the rotation-invariance and rotation-equivariance of images.

In Figure 6, we analyzed several image samples by rotating them and examining the patterns of their equivariance scores. For objects that are equivariant to rotation, such as cars and birds, the equivariance scores exhibited periodic and regular patterns in response to rotation. In contrast, images that are rotation-invariant, like flowers, generated noisy equivariance scores, highlighting the distinction between the two types.

To verify whether performance drops on rotation-invariant datasets, we measured the linear classification accuracy of the pretrained backbone on other datasets across four discrete 90-degree orientations. As shown in Table 4 in Appendix B.2, the GIE model outperformed the baseline backbones, such as the E(2)-CNN and the E(2)-CNN with group pooling, even on rotation-invariant datasets like Oxford 102 Flowers and MTARSI (Wu et al., 2020). This result indicates that the equivariance score not only represents recognizable orientations but also functions as a complex attention weight, supporting the robustness of GIE.

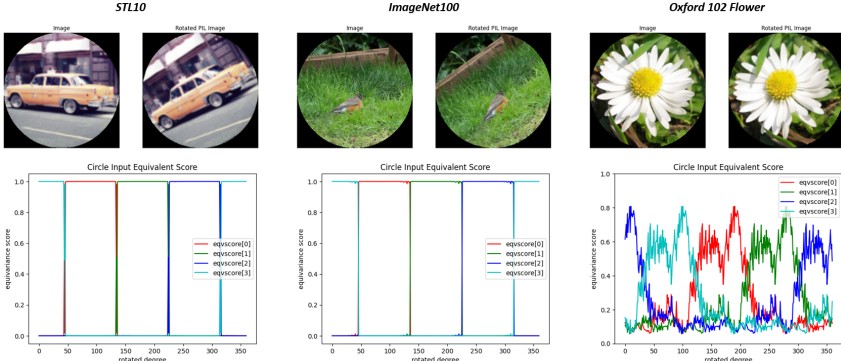

Figure 6: Image samples and corresponding equivariance score graphs across rotated degrees. Additional equivariance score results are illustrated in Figure 13.

**Semantic segmentation using Pascal VOC datasets** To verify whether the GIE model could also produce strong rotation-invariant features for dense prediction tasks, we applied it to the semantic segmentation task using the Pascal VOC (Everingham et al., 2010) dataset. The results showed that the GIE model backbone outperformed other baseline models in terms of mIOU and Pixel Accuracy at all angles (0, 90, 180, 270 degrees). Details of the experimental setup and results can be found in Appendix B.7.

**Extension on $p8$-group** Extending the GIE concept to a group of order $N$ is a natural progression. We conducted experiments on CIFAR10 to apply the GIE method to the $p8$-group. Since mathematical exact equivariance is not feasible due to bilinear interpolation occurring at the pixel level when the image is rotated by 45 degrees, we employed a specialized augmentation method to address this issue. The details of the procedure and results can be found in Appendix B.8.

## 5 LIMITATIONS AND FUTURE WORK

The GIE model relies on the rotation-equivariant properties of the features. Therefore, a GCNN backbone must be used, and detailed input image size settings are required to ensure exact equivariance. Additionally, a loss function is required to learn the equivariance score. Currently, the orientation loss is only used within the contrastive learning framework, but we plan to extend its applicability to supervised learning in future work. Furthermore, we will also more explore extending the method to a rotation group of order $N$ and applying it to tasks such as image segmentation.

## ETHICS STATEMENT

This research adheres to ethical standards in AI, including considerations for data use, fairness, and privacy. We utilized publicly available datasets (CIFAR10, STL10, ImageNet100, Pascal VOC, etc.) that do not contain personally identifiable information. While our models may inherit biases present in these datasets, we did not intentionally introduce or analyze biases, and we advocate responsible use of our methods to minimize potential misuse. Our research does not involve human subjects or require Institutional Review Board (IRB) approval, and all methodologies, results, and models have been transparently documented to maintain integrity.

## REPRODUCIBILITY STATEMENT

All experiments in this study have been thoroughly documented to ensure reproducibility. The datasets used (CIFAR10, STL10, ImageNet100, Pascal VOC, etc.) are publicly available, and the code and model configurations have been clearly specified. All algorithms and hyperparameter settings described in this paper are detailed explicitly, and the code will be made publicly available.

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

## A    Supplementary Information Regarding GIE

### A.1    Preliminaries on GCNN

**Group convolution**   Group convolution is a generalization of traditional convolution in neural networks, where the transformation group (such as rotations, reflections, or translations) is applied to the feature maps. Instead of performing convolution only over spatial translations, group convolution processes data using symmetries from a specific group. This allows the model to capture patterns and symmetries beyond simple shifts, making it more robust to transformations like rotations.

**Equivariance and invariance**   Equivariance and invariance are two important concepts in representation learning. Equivariance refers to a property where a transformation applied to the input results in a corresponding transformation in the output. For example, a neural network is equivariant if rotating an image leads to a rotated feature map in the output. Formally, for a function $f$ and a transformation $T$, the function is equivariant if $f(T(x)) = T(f(x))$. On the other hand, invariance means that the output remains unchanged when a transformation is applied to the input. A function is invariant to a transformation $T$ if $f(T(x)) = f(x)$. Invariance is useful for tasks where the output should be insensitive to specific transformations, like object recognition regardless of orientation, while equivariance is crucial for capturing structured changes in the input data.

**Exact equivariance**   Exact equivariance refers to a strict form of equivariance, where the model's output perfectly follows the transformation applied to the input. In an exactly equivariant system, the transformation of the input always leads to a predictable and mathematically precise transformation of the output, without any loss of information. This differs from approximate equivariance, where the correspondence between transformed inputs and outputs may not be perfect but is close enough for practical purposes.

### A.2    Pytorch-style pseudocode for guiding invariance

The pseudocode for guiding invariance discussed in Section 3.4 is presented in Algorithm 1.

**Algorithm 1** Pytorch-style pseudocode for guiding invariance process under the rotation group of order $N$.

```python
# Input: F(X), S(X), N  # Feature representation, equivariant score, and
    order N
# Output: H(X)  # Output tensor

def H(F_X, S_X, N):
    # Step 1: Assign equivariant score
    eqv_score = S_X

    # Step 2: Assign feature representation
    feature_repr = F_X

    # Step 3: Generate permuted representations (order N case)
    permuted_reprs = [torch.roll(feature_repr, shifts=i, dims=-1) for i
        in range(N)]
    permuted_reprs = torch.stack(permuted_reprs, dim=-1)

    # Step 4: Perform weighted sum of permuted representations
    H_X = torch.matmul(permuted_reprs, eqv_score.unsqueeze(dim=-1)).
        squeeze(dim=-1)

    # Output result H(X)
    return H_X
```

## B    DETAILED INFORMATION AND ADDITIONAL INSIGHTS REGARDING THE EXPERIMENTS

### B.1    DETAILS OF EXPERIMENTS

**CIFAR10**    CIFAR10 is an image recognition dataset consisting of 60,000 32x32 color images across 10 object classes. Each class contains 6,000 images, with 5,000 designated for training and 1,000 for testing. We utilized all 50,000 training images for self-supervised pretraining and subsequently evaluated linear classification accuracy by attaching a linear classifier to the pretrained backbone. The evaluation was performed using the full set of 50,000 training images and 10,000 test images. For the exact equivariance of feature, we set the training image size 33x33 in the experiments.

We used an E(2)-CNN backbone following the ResNet18 architecture. The initial number of channels consisted of 20 regular representation units (80 dimensions), and the final output feature increased by a factor of 8 to become 160-regular representation units (640 dimensions). The equivariance predictor uses two $1 \times 1$ group-equivariant convolution layers, employs 512-regular representation units (2048 dimensions) for the intermediate node type, and, as previously mentioned, uses 1-regular representation unit (4 dimensions) for the output type. Both SimCLR and SimSiam utilized the SGD (Ruder, 2016) optimizer, with a learning rate of 0.06 and a batch size of 512. We conducted SSL training for 800 epochs, after which the trained backbone was frozen, and a linear classifier was attached for 100 epochs to measure linear classification accuracy. To prevent overfitting, the base learning rate was set to 30 and decreased using a cosine learning rate scheduler.

**STL10**    STL10 is an image recognition dataset specifically designed for unsupervised and semi-supervised learning. It includes 10 classes and features 100,000 unlabeled images, 5,000 training images, and 8,000 validation images. We conducted pretraining on a combined set of 105,000 images, incorporating both the unlabeled and training subsets, and then performed linear evaluation using 5,000 training images and 8,000 validation images. For the exact equivariance of feature, we set the training image size 97x97 in the experiments.

We used an E(2)-CNN backbone following the ResNet18 architecture. The initial number of channels consisted of 39-regular representation units (156 dimensions), and the final output feature increased by a factor of 8 to become 312-regular representation units (1248 dimensions). The equiv-

ariance predictor uses three $1 \times 1$ group-equivariant convolution layers, employs 512-regular representation units (2048 dimensions) for the intermediate node type, and, as previously mentioned, uses 1-regular representation unit (4 dimensions) for the output type. For SimCLR training, we used a learning rate of 0.6, a batch size of 512, 400 epochs, and the LARS (You et al., 2017) optimizer. For SimSiam training, we used a learning rate of 0.1, a batch size of 512, 400 epochs, and the SGD optimizer. To measure linear classification accuracy, we froze the trained backbone and attached a linear classifier, which was trained for 100 epochs with a learning rate of 1.0.

**ImageNet100** ImageNet (Russakovsky et al., 2015) is a large-scale image recognition dataset comprising approximately 1,280,000 images. For our experiments, we used a subset, ImageNet100, which includes 100 selected classes as described in Tian et al. (2020). We performed pretraining using the training data, followed by linear evaluation on both the training and validation datasets. For the exact equivariance of feature, we set the image size 225x225 in the experiments.

We used an E(2)-CNN backbone following the ResNet18 architecture. The initial number of channels consisted of 39-regular representation units (156 dimensions), and the final output feature increased by a factor of 8 to become 312-regular representation units (1248 dimensions). The equivariance predictor uses three $1 \times 1$ group-equivariant convolution layers, employs 512-regular representation units (2048 dimensions) for the intermediate node type, and, as previously mentioned, uses 1-regular representation unit (4 dimensions) for the output type. For SimCLR training, we used a learning rate of 0.3, a batch size of 256, 400 epochs, and the LARS optimizer. For SimSiam training, we used a learning rate of 0.05, a batch size of 256, 400 epochs, and the SGD optimizer. To measure linear classification accuracy, we froze the trained backbone and attached a linear classifier, which was trained for 100 epochs with a learning rate of 1.0.

### B.2 LINEAR CLASSIFICATION ACCURACY FOR OTHER DATASETS

To verify the transferability performance of the model, we conducted evaluation experiments measuring linear classification accuracy on various natural image datasets, including STL10, Stanford Cars (Krause et al., 2013), Caltech256 (Griffin et al., 2007), FGVC-Aircraft (Maji et al., 2013), CUB-200-2011 (Wah et al., 2011), as well as rotation-invariant datasets like Oxford 102 Flowers and MTARSI, using an 18-depth E(2)-CNN model trained on the ImageNet100 dataset. As shown in Table 4, our E(2)-CNN GIE model exhibited the highest performance across all categories of the datasets.

Table 4: Linear classification accuracy for other datasets. We measured the linear classification accuracy of the E(2)-CNN backbones with 18-depth, pretrained on ImageNet100, across four discrete 90-degree rotations for other datasets.

| Dataset | E(2)-CNN | E(2)-CNN Gpool | E(2)-CNN GIE(ours) |
|---|---|---|---|
| STL10 | 83.11 | 84.48 | **87.89** |
| Stanford Cars | 22.80 | 20.82 | **32.39** |
| Caltech256 | 57.25 | 59.16 | **64.11** |
| FGVC-Aircraft | 29.04 | 26.37 | **38.94** |
| CUB-200-2011 | 22.04 | 20.98 | **25.45** |
| Oxford 102 Flowers | 85.33 | 83.22 | **86.06** |
| MTARSI | 85.92 | 82.44 | **87.46** |

### B.3 COMPARISON OF GROUP ALIGNING(SHIFT) AND GROUP ATTENTIONING(SOFTMAX)

As another guiding invariance method, we employed group aligning operations. Group aligning was introduced in Lee et al. (2023) to learn rotation-invariant descriptors for visual correspondence tasks. In that paper, an orientation map is extracted and a cyclic shift is performed to the dominant orientation dimension to ensure the rotation invariance of the descriptors. Similarly, we extracted the dominant dimension, which has the maximum value, from our learned equivariance score $S(X)$ and performed group aligning on the rotation equivariant feature $F(X)$ by cyclically shifting it to

this dimension. Formally, we can define $H(X)$ with the following expression:

$$H(X) := F(r^{-k}X), \quad \text{where } k = argmax(S(X)) \tag{11}$$

We additionally experimented with group aligning as a guiding invariance process to assess any performance differences compared to group attentioning. Table 5 organizes the SimCLR and Sim-Siam performance for STL10 and ImageNet100 according to the guiding invariance process (GIP). 'None' represents the performance without using GIE, 'Align' represents the use of group align-ing, and 'Attention' represents the use of group attentioning as the GIP. STL10 used the E(2)-CNN model with 18-depth, while ImageNet100 used the E(2)-CNN with 50-depth. For the non-rotated dataset, the performance of $F(X)$ is reported, and for the rotated dataset, the performance of $H(X)$ is reported.

Table 5: Ablation on guiding invariance process. For each SSL method, boldface highlights the best performance among the experiments in each GIP.

| SSL method | GIP | STL10 | STL10-R | ImageNet100 | ImageNet100-R |
|---|---|---|---|---|---|
| SimCLR | None | 85.50 | 80.95 | 80.58 | 76.82 |
| | Align | 86.33 | 86.16 | **81.48** | **81.37** |
| | Attention | **86.45** | **86.19** | 81.34 | 81.05 |
| SimSiam | None | 86.01 | 84.15 | 73.12 | 71.96 |
| | Align | 87.13 | 88.10 | 75.32 | 76.41 |
| | Attention | **87.48** | **88.31** | **75.66** | **76.46** |

In the experiments, while group attentioning generally outperformed group aligning, exceptions oc-curred in the SimCLR ImageNet100 experiment. Also, the performance difference between group aligning and group attentioning was not significant, even when group attentioning was higher. Ad-ditionally, regardless of whether aligning or attentioning was used, performance was higher in both non-rotated and rotated cases compared to when the GIE methodology was not used at all. There-fore, the results show that either group aligning or group attentioning can yield good performance in our GIE method.

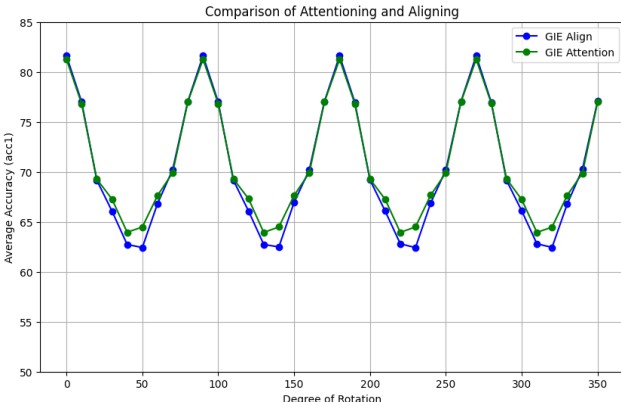

Figure 7: Results of STL10 SimCLR pretrained models for 10° rotated inference.

However, the group attention method has an advantage in *smoothness* over the align method. Fig-ure 7 presents the results of a graph drawn after training a linear classifier with random rotation augmentation on the STL10 dataset and conducting inference on a dataset rotated in 10-degree in-crements. Measuring performance in 10-degree increments reveals that while the performance of group attentioning and group aligning methods is similar around 90 degrees, group attentioning out-performs group aligning around 45 degrees. This difference can be attributed to the fact that the features created by group attentioning are continuous with respect to rotation, whereas those from group aligning are discrete.

## B.4  ABLATION STUDY OF $\beta$ (ON STL10)

Table 6: Ablation study of $\beta$ on STL10 dataset.

| Model | $\beta = 0.1$ | | $\beta = 0.2$ | | $\beta = 0.3$ | | $\beta = 0.4$ | |
|---|---|---|---|---|---|---|---|---|
| | NR | R | NR | R | NR | R | NR | R |
| SimCLR E(2)-CNN GIE $F(X)$ | 86.45 | 82.43 | 86.75 | 84.30 | 86.50 | 82.61 | 86.04 | 82.58 |
| SimCLR E(2)-CNN GIE $H(X)$ | 86.30 | 86.19 | 87.11 | 86.71 | 86.29 | 86.07 | 85.75 | 85.70 |

As shown in Table 6, we generally select $\beta$ between 0.1 and 0.4, as this range tends to yield good performance. Therefore, we did not engage in overly sensitive tuning for specific datasets. Additional experiments on SimCLR with the STL10 dataset show that beta achieves the highest performance at 0.2. However, since this value can depend on the type of data and experimental settings, we consistently set the beta value to 0.1 for all our experiments to maintain uniformity, as there was no significant performance difference with varying beta values.

## B.5  ABLATION STUDY OF INITIAL CHANNELS (ON CIFAR10)

Table 7: Comparison of GIE models with different base widths.

| | GIE-16 | GIE-20 | GIE-24 |
|---|---|---|---|
| Initial Channels | 16 | 20 | 24 |
| GPU Memory (GB) | 7.6 | 8.6 | 10.5 |
| Encoder Params (M) | 2.14 | 3.26 | 4.62 |
| NR | 91.34 | 91.72 | 92.52 |
| R | 91.27 | 92.01 | 92.55 |

We examined how the performance of the GIE model changes as the initial channels of the E(2)-CNN vary. As shown in Table 7, we reported GPU memory usage, encoder parameters, and performance on both the NR and R datasets from CIFAR10 when the initial number of channels was set to 16, 20, and 24. As observed, increasing the initial number of channels results in higher GPU memory consumption and more encoder parameters, which in turn improves the performance on both the NR and R datasets. In the default GIE setting for CIFAR10, the initial number of channels was set to 20 to strike an appropriate trade-off between GPU memory usage and performance.

## B.6  ABLATION STUDY OF 15, 30, 45 AUGMENTATION DEGREE (ON STL10)

We conducted additional experiments using random rotation augmentation with smaller ranges of -15 to 15 degrees and -30 to 30 degrees. These comparisons aim to illustrate why the -45 to 45 degree range is more suitable for evaluating rotation invariance when utilizing the E(2)-CNN backbone. As shown in Figure 8a, while methods with less random rotation may perform better at the 0-degree point, the approach using a range of -45 to 45 degrees demonstrates greater stability across all angles, thereby confirming its suitability for evaluating rotation invariance in the E(2)-CNN backbone.

Furthermore, we performed additional experiments comparing the results of transformations with and without circular cropping. As illustrated in Figures 8b, 8c, and 8d, the application of circular cropping across all random rotation augmentations results in significantly greater stability and superior performance at all angles. Therefore, we can conclude that the use of circular cropping improves overall performance for evaluating rotation invariance.

## B.7  EXPERIMENTAL DETAILS AND RESULTS FOR PASCAL VOC SEGMENTATION

We used an image encoder pretrained on ImageNet100 as the backbone, removing the global average pooling layer to preserve the final feature size. A simple segmentation head consisting of two 1x1 convolution layers was attached to the image encoder, followed by bilinear upsampling to restore the original image size. With the backbone frozen, we trained the model for 20 epochs using

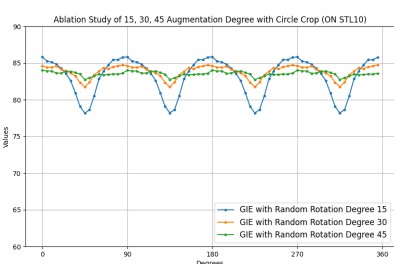

(a) Comparison of model performance under random rotations of 15°, 30°, and 45°

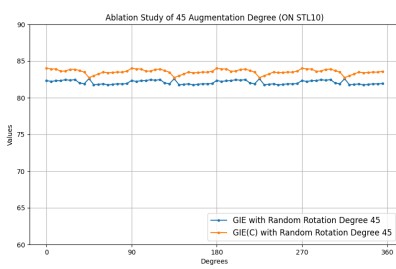

(b) Comparison of model performance with and without circle crop under 45° random rotation

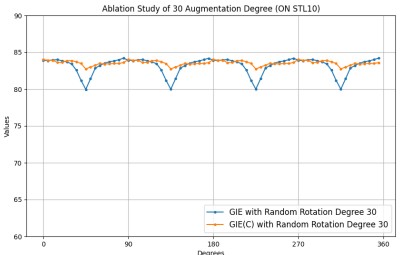

(c) Comparison of model performance with and without circle crop under 30° random rotation

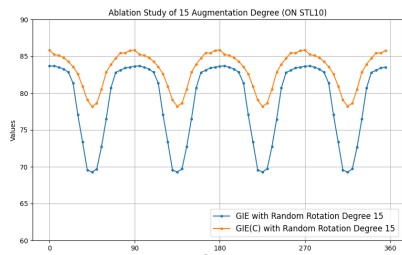

(d) Comparison of model performance with and without circle crop under 15° random rotation

Figure 8: Ablation study of 15, 30, 45 augmentation degree (on STL10). "GIE" denotes our model that does not employ circle crop, whereas "GIE(C)" signifies our model that incorporates circle crop.

Table 8: Evaluation results for Pascal VOC segmentation. We performed Pascal VOC segmentation using models pretrained on ImageNet100. The experiments were divided into two settings: training with only 0-degree images and training with images rotated by 0, 90, 180, and 270 degrees. The results highlight the best mIOU and Pixel Accuracy values for each angle in bold.

| Trained Degree | Model | Mean IOU | | | | Pixel Accuracy | | | |
|---|---|---|---|---|---|---|---|---|---|
| | | 0 degree | 90 degree | 180 degree | 270 degree | 0 degree | 90 degree | 180 degree | 270 degree |
| 0 | ResNet50 | $0.1672 \pm 0.0014$ | $0.0754 \pm 0.0013$ | $0.0847 \pm 0.0013$ | $0.0752 \pm 0.0010$ | $0.8240 \pm 0.0008$ | $0.7653 \pm 0.0020$ | $0.7714 \pm 0.0015$ | $0.7654 \pm 0.0019$ |
| | ResNet50(R) | $0.1416 \pm 0.0017$ | $0.1428 \pm 0.0012$ | $0.1423 \pm 0.0022$ | $0.1433 \pm 0.0022$ | $0.8097 \pm 0.0007$ | $0.8115 \pm 0.0003$ | $0.8107 \pm 0.0008$ | $0.8117 \pm 0.0007$ |
| | E(2)-CNN | $0.1840 \pm 0.0011$ | $0.0799 \pm 0.0016$ | $0.1052 \pm 0.0036$ | $0.0797 \pm 0.0016$ | $0.8236 \pm 0.0019$ | $0.7682 \pm 0.0021$ | $0.7838 \pm 0.0011$ | $0.7670 \pm 0.0019$ |
| | E(2)-CNN Gpool | $0.1831 \pm 0.0034$ | $0.1831 \pm 0.0035$ | $0.1831 \pm 0.0034$ | $0.1831 \pm 0.0034$ | $0.8173 \pm 0.0014$ | $0.8173 \pm 0.0014$ | $0.8173 \pm 0.0014$ | $0.8173 \pm 0.0014$ |
| | E(2)-CNN GIE(ours) | $\mathbf{0.1884 \pm 0.0031}$ | $\mathbf{0.1884 \pm 0.0031}$ | $\mathbf{0.1884 \pm 0.0032}$ | $\mathbf{0.1884 \pm 0.0031}$ | $\mathbf{0.8281 \pm 0.0010}$ | $\mathbf{0.8281 \pm 0.0010}$ | $\mathbf{0.8281 \pm 0.0010}$ | $\mathbf{0.8281 \pm 0.0010}$ |
| 0, 90, 180, 270 | ResNet50 | $0.1491 \pm 0.0030$ | $0.1095 \pm 0.0027$ | $0.1106 \pm 0.0022$ | $0.1064 \pm 0.0021$ | $0.8135 \pm 0.0013$ | $0.7927 \pm 0.0011$ | $0.7920 \pm 0.0010$ | $0.7907 \pm 0.0011$ |
| | ResNet50(R) | $0.1429 \pm 0.0012$ | $0.1428 \pm 0.0019$ | $0.1436 \pm 0.0017$ | $0.1434 \pm 0.0025$ | $0.8102 \pm 0.0009$ | $0.8116 \pm 0.0005$ | $0.8113 \pm 0.0010$ | $0.8118 \pm 0.0008$ |
| | E(2)-CNN | $0.1687 \pm 0.0021$ | $0.1693 \pm 0.0031$ | $0.1687 \pm 0.0031$ | $0.1688 \pm 0.0021$ | $0.8142 \pm 0.0018$ | $0.8143 \pm 0.0019$ | $0.8141 \pm 0.0019$ | $0.8145 \pm 0.0016$ |
| | E(2)-CNN Gpool | $0.1825 \pm 0.0035$ | $0.1825 \pm 0.0035$ | $0.1825 \pm 0.0035$ | $0.1825 \pm 0.0035$ | $0.8174 \pm 0.0013$ | $0.8174 \pm 0.0013$ | $0.8174 \pm 0.0013$ | $0.8174 \pm 0.0013$ |
| | E(2)-CNN GIE(ours) | $\mathbf{0.1884 \pm 0.0031}$ | $\mathbf{0.1884 \pm 0.0031}$ | $\mathbf{0.1884 \pm 0.0031}$ | $\mathbf{0.1884 \pm 0.0031}$ | $\mathbf{0.8281 \pm 0.0008}$ | $\mathbf{0.8281 \pm 0.0008}$ | $\mathbf{0.8281 \pm 0.0008}$ | $\mathbf{0.8281 \pm 0.0008}$ |

cross-entropy loss and reported the mean Intersection over Union (mIOU) and Pixel Accuracy. We repeated the same experiment five times and calculated the mean and standard deviation.

As shown in Table 8, the results indicate that the GIE model achieved the same mIOU and Pixel Accuracy on images rotated by 90, 180, and 270 degrees as it did on the original images, outperforming other baseline models. Furthermore, when trained on data rotated by 90, 180, and 270 degrees, other models exhibited a performance drop on the original images, whereas the GIE model maintained its performance, demonstrating that the performance gap could not be closed.

### B.8 EXTENSION ON $p8$-GROUP

We conducted experiments on CIFAR10 to apply the GIE method to the $p8$-group. When an image is rotated by 45 degrees, bilinear interpolation occurs at the pixel level, making exact mathematical equivariance impossible. Therefore, we doubled the dataset size by adding 45-degree rotated images to the original dataset and used a rotation augmentation transform in the range of [-22.5, 22.5] degrees for random rotation during training. An important point here is that, during contrastive

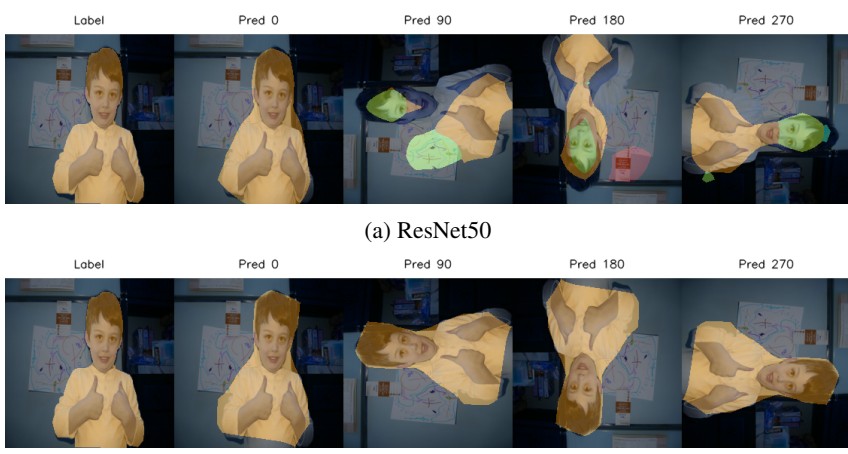

(a) ResNet50

(b) E(2)-CNN GIE (ours)

Figure 9: Semantic segmentation sample results of (a) baseline SimCLR-trained ResNet50 backbone and (b) our E(2)-CNN GIE-trained backbone

Table 9: Comparison of CIFAR10 experiments using $p4$-group and $p8$-group GIE.

|  | $p4$-**group GIE** | $p8$-**group GIE** |
| --- | --- | --- |
| **GPU Memory (GB)** | 8.5 | 8.5 |
| **Initial Channels** | 20 | 10 |
| **Encoder Params (M)** | 3.26 | 1.8 |
| **(0,5,10,…,355) R** | $86.750 \pm 0.177$ | $87.087 \pm 0.530$ |

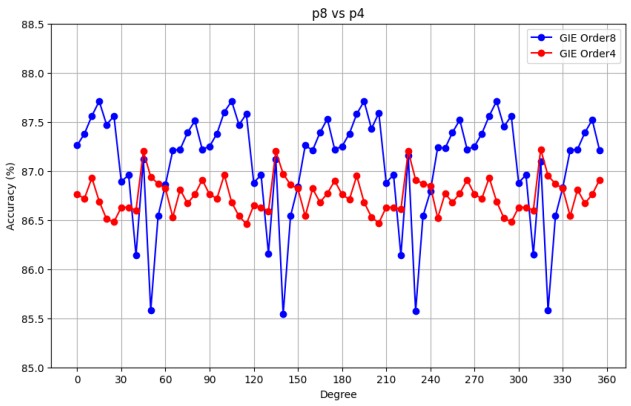

Figure 10: Comparison of $p4$-group GIE and $p8$-group GIE.

learning, images from the original dataset form positive pairs only with each other, while 45-degree rotated images form positive pairs only with each other. In other words, images from the 0-degree and 45-degree datasets are not mixed for positive pair sampling. This setup encourages the 8-dimensional equivariance score to represent distinct values for images at 0 degrees and those rotated by 45 degrees.

We conducted the same evaluation for random rotation degrees as was done for the $p4$-group. Additionally, to facilitate a fair comparison with the $p4$-group GIE, we reduced the initial number of channels to 10 during training to ensure similar GPU memory consumption. As shown in Table 9 and Figure 10, the $p8$-group GIE achieved a higher mean accuracy with fewer encoder parameters. However, it exhibited a higher standard deviation, indicating unstable performance. This instability may be attributed to the suboptimal augmentation method used to address the lack of exact equivari-

ance at 0 and 45 degrees. Nevertheless, the $p8$-group GIE demonstrated better efficiency compared to the $p4$-group GIE, indicating potential for further improvement in future research.

# C  ADDITIONAL VISUALIZATIONS

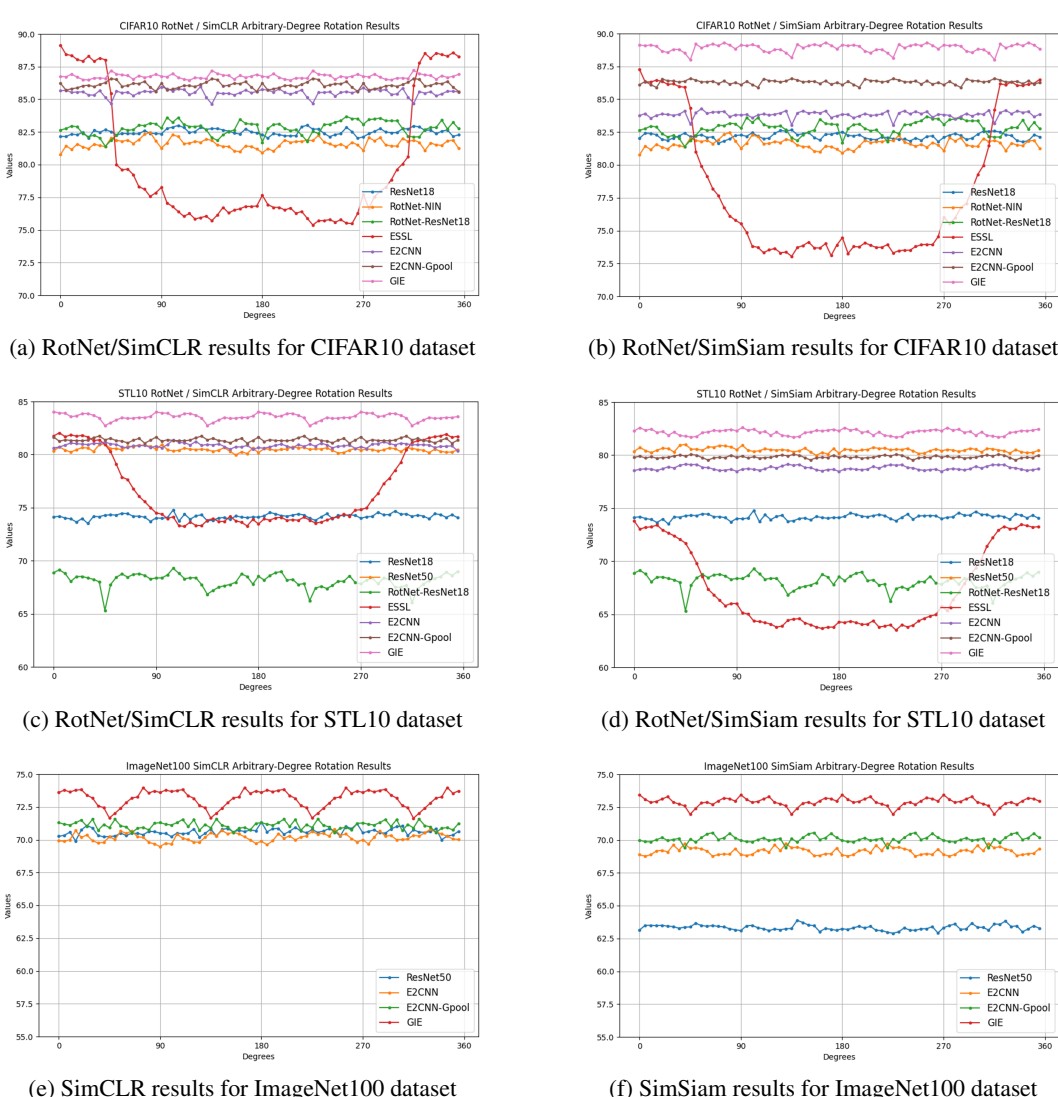

(a) RotNet/SimCLR results for CIFAR10 dataset

(b) RotNet/SimSiam results for CIFAR10 dataset

(c) RotNet/SimCLR results for STL10 dataset

(d) RotNet/SimSiam results for STL10 dataset

(e) SimCLR results for ImageNet100 dataset

(f) SimSiam results for ImageNet100 dataset

Figure 11: Results of arbitrary-degree rotations.

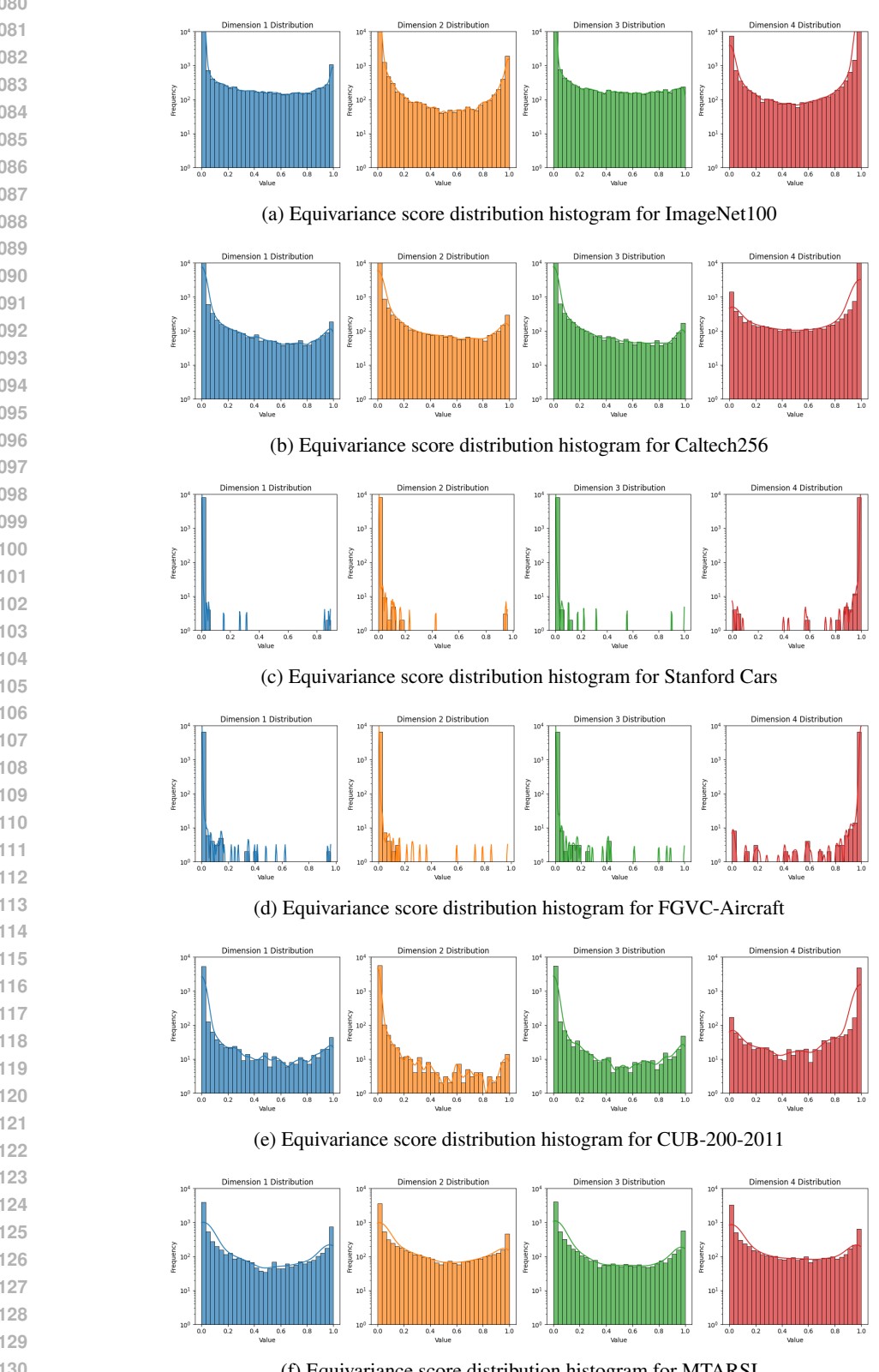

Figure 12: Equivariance score distribution histograms for different datasets. (a) ImageNet100, (b) Caltech256, (c) Stanford Cars, (d) FGVC-Aircraft, (e) CUB-200-2011, (f) MTARSI

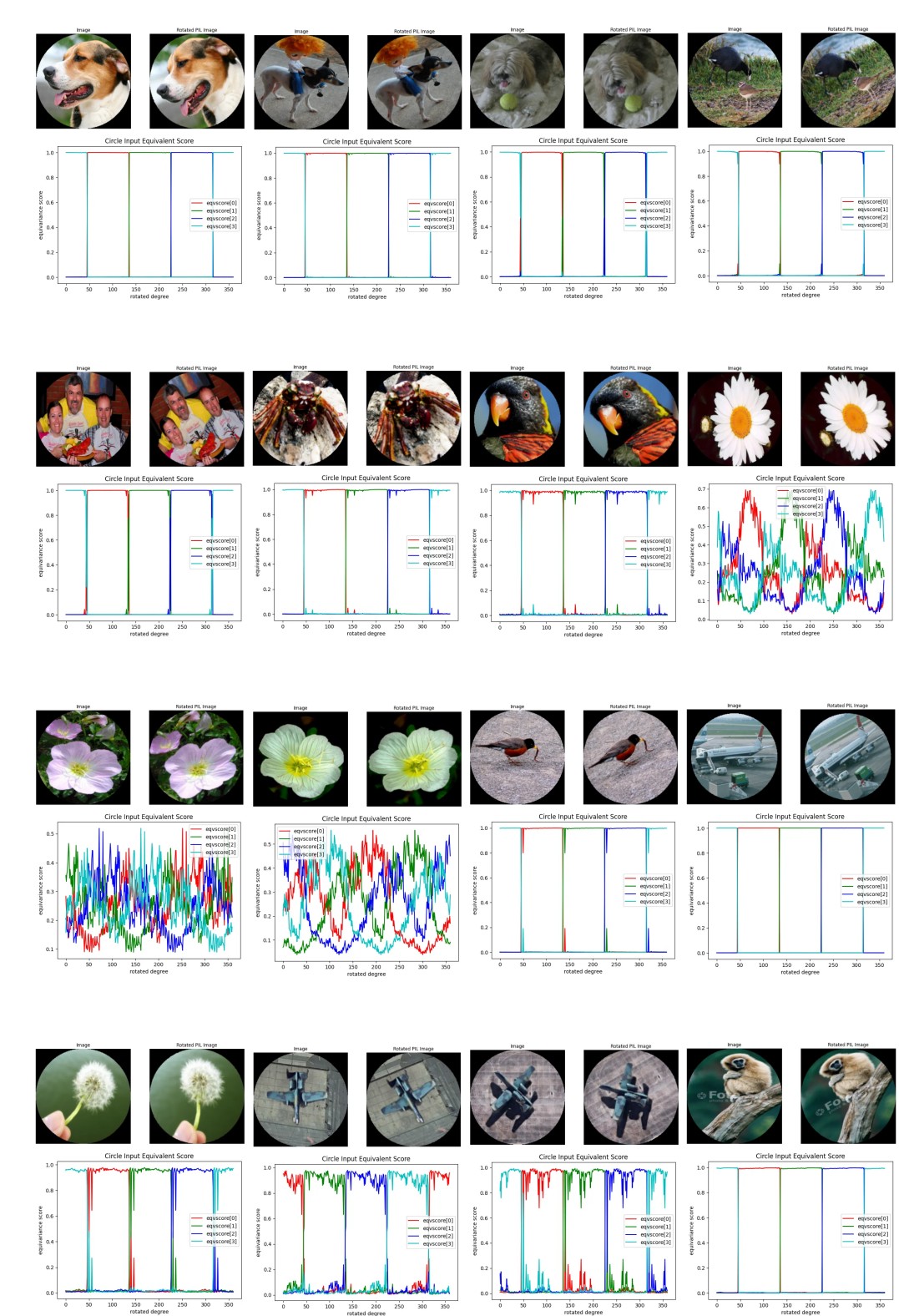

Figure 13: Examples of equivariance score samples.

