# OpenReview forum: "Learning Rotation-Invariant Representation using Rotation-Equivariant CNNs"
_ICLR.cc/2025/Conference — ICLR 2025 Conference Withdrawn Submission_

### Official Review · Reviewer_5JN2 · 2024-11-04

**Soundness:** 2
**Presentation:** 3
**Contribution:** 2
**Rating:** 3
**Confidence:** 4

**Summary:**

The paper proposes to learn rotation equivariant features for self-supervised training. This is done using a rotation-equivariant backbone network built from e2cnn library. It uses an equivariant predictor that provides the equivariance score guides the rotation-equivariant features.

**Strengths:**

The paper learns rotation invariant features which are helpful for rotation invariant datasets.
The approach is simple and easy to follow.

**Weaknesses:**

1. The technical novelty is limited as the authors have used existing rotation equivariant network and trained it to generate rotation equivariant only.
2. The approach has high requirements. It requires specific CNN backbone network. It is not compatible with Vision Transformers.
3. The approach is compared with just a few baselines: SIMCLR and SimSiam.
4. The performance of the approach is lower in some cases when dataset is not rotated. This shows its limited use cases (only for rotation invariant datasets).
5. Basic RotNet approach is not compared with the E(2)-CNN.

**Questions:**

1. Comparison with newer self-supervised approaches like MoCo etc.
2. Can it be used with Vision Transformers?
3. The results are quite close to each other. Having a mean and standard deviation plot will strengthen that increase is because of the method and not due to randomness.

---

### Official Review · Reviewer_o1fp · 2024-11-04

**Soundness:** 2
**Presentation:** 3
**Contribution:** 2
**Rating:** 5
**Confidence:** 4

**Summary:**

This paper integrates rotation-equivariant CNNs (E2CNNs) into self-supervised learning (SSL) methods and introduces the Guiding Invariance with Equivariance (GIE) training approach. GIE leverages rotation-equivariant CNNs to produce both equivariant features and scores, which are combined to yield rotation-invariant representations. Experimental results show that SSL+GIE+E2CNN consistently outperforms SSL with E2CNN alone (or non-equivariant networks), demonstrating the effectiveness of this approach.

**Strengths:**

This paper aims to enhance the robustness of self-supervised learning (SSL) methods to rotation by leveraging rotation-equivariant CNNs. This approach raises an intriguing question: is the proposed GIE method superior to simply combining rotation-equivariant CNNs with rotation-based data augmentation? Given the discrete nature of digital images, achieving exact rotation equivariance is challenging. Previous work on rotation-equivariant networks has shown that adding rotation augmentation can still improve accuracy, despite the inherent rotation equivariance of these networks. Interestingly, this paper demonstrates that E2CNN with GIE outperforms E2CNN with rotation augmentation in rotation-invariant classification tasks, suggesting that SSL+GIE achieves a level of robustness that data augmentation alone cannot. Depending on further evidence, this could be viewed as either a strength or a potential weakness.

**Weaknesses:**

1. Effectiveness of SSL and GIE with E2CNN: A primary concern is that an E2CNN can potentially be trained without SSL or GIE and still achieve robust rotation invariance. If this is the case, the need for incorporating SSL and GIE to improve E2CNN becomes unclear. While Tables 1 and 2 show that SSL+GIE+E2CNN outperforms SSL+E2CNN, it’s worth noting that E2CNN is inherently equivariant to the p4-group, which may indicate a limitation in the number of orientation samples. The experiment in the appendix, comparing the p4-group and p8-group, raises questions, as the authors have matched GPU memory usage instead of parameter count—an approach that deviates from common practice in equivariant network studies, which typically prioritize matching parameter counts. Since matched parameter counts are more relevant for fair comparisons, this paper should provide evidence that SSL+GIE+E2CNN is superior to SSL+E2CNN, and to E2CNN alone, with matched parameter counts for p16- and even p32-groups. I think that matched parameter count is more important than matched GPU memory consumption in the long run as GPU hardware is evolving fast.

I tend to change score based on the answer to this question.


2. Motivation and Benchmark Choices: Some motivations in Figure 1 are debatable. While humans may struggle with reading rotated text, recognizing rotated objects generally relies more on shape than orientation, so rotation is less problematic for object recognition without canonicalization. Furthermore, E2CNN has achieved over 99% accuracy on rotated handwritten digits in MNIST, effectively distinguishing between rotated 6 and 9, likely due to shape differences inherent in handwriting styles. Despite the relevance of rotated MNIST as a benchmark in rotation-equivariant literature, this dataset is absent from the paper’s evaluations. MNIST offers a zero-background advantage, which is ideal for testing rotation without cropping, so including it could strengthen the results and make us easy to compare the results in equivariant literature.


3. Parameter Comparison in Results: The SSL+GIE+E2CNN models consistently have more parameters than SSL+E2CNN due to the additional equivariance predictor, which makes the results less convincing. For instance, in Table 1, SimSiam+GIE+E2CNN+EqvPred (76.54%) uses 12.83M parameters, while SimSiam+GIE+E2CNN+group pooling (73.26%) has 11.14M parameters. This additional 15% in parameters only yields a 4% accuracy increase, which is not particularly impressive. It would strengthen the paper if the authors could demonstrate that SSL+GIE+E2CNN performs better than SSL+E2CNN with fewer or equal parameters.

**Questions:**

1. In the bottom diagram of Figure 1, the pipeline includes an equivariant encoder, GIE, and a classifier. However, lines 315-316 state, “After pretraining, we froze the pretrained backbone and attached a linear classifier to measure linear classification accuracy.” Since E2CNN can serve as the backbone, does this mean that it is followed by linear layers without GIE during testing? This is somewhat unclear; could you please clarify?
2. In Figure 4, the E2CNN-Gpool curves (brown) appear flatter than the GIE curves (pink), although they are lower overall. How is robustness quantified here? Can it be inferred that curves with lower variance are more robust than those with higher variance?

---

### Official Review · Reviewer_TuRg · 2024-11-04

**Soundness:** 3
**Presentation:** 2
**Contribution:** 2
**Rating:** 3
**Confidence:** 2

**Summary:**

The paper presents an approach for developing rotation-invariant features in self-supervised learning (SSL) by leveraging rotation-equivariant CNNs, which is called as “Guiding Invariance with Equivariance” (GIE) method. The GIE method improves the rotation invariance of SSL frameworks like SimCLR and SimSiam, by using an equivariance score to guide rotation-equivariant features via an attention-weighted mechanism. The approach demonstrates improved performance in handling both fixed and arbitrary-degree rotations, with experiments on CIFAR10, STL10, and ImageNet100 datasets highlighting its robustness across various degrees of rotation.

**Strengths:**

- The proposed method is conceptually strong and well motivated to addres  a notable limitation in SSL methods.
- Experiments show that the proposed method can achieve higher classification accuracy for the rotated datasets.

**Weaknesses:**

- The GIE method relies on rotation-equivariant properties, requiring GCNN backbones, which may limit the method’s applicability in scenarios where GCNNs are impractical.
- While the study includes datasets with varying rotation properties, a broader analysis of the model’s effectiveness on datasets with more complex real-world invariances could strengthen its claims.
- The experiments are mainly restricted to classifications. It would be better validate the method on the applications with rotation variance issues, such as object detection and key point matching.
- Following the last point, the image level global rotation is not very common in real-world scenarios. So the experiments on imag classification cannot fully demonstrate the benefits of the method.
As shown in the experiments, the proposed method cannot bring improvement for the original dataset NR dataset. It implies the image-level rotation is not common in the real world.
Differently, the rotation issue can be common in the local features. And the rotation invariant representation can be validated better on these related applications.

**Questions:**

Please check the weakness points and address them.

---

### Official Review · Reviewer_8QTW · 2024-11-04

**Soundness:** 2
**Presentation:** 2
**Contribution:** 2
**Rating:** 3
**Confidence:** 4

**Summary:**

This work presents GIE, a method able to learn rotation invariant representations based on equivariant neural networks. The authors experimentally show that GIE extracts invariant features across the 90 rotations considered in the p4 group used.

**Strengths:**

The paper is clearly written and well-structured.

**Weaknesses:**

The main weakness of this work is that, after reading the paper multiple times, it is still unclear to me what the contribution of the paper is over existing work. I will outline this in detail as follows:

- **positioning relative to existing work.** As mentioned in the related work section, there already exists multiple works that learn robust equivariant and invariant representations in an unsupervised manner. However, the authors do not outline what the differences are between this work and existing works. The objective of the related work section should be to position your work relative to others. Not to list what exists.

  To complement my previous statement, papers such as [1], which propose a very similar method to learn invariant and equivariant representations, are not cited, let alone discussed. In fact, the formulation of [1] is more general, and considered wider, more complex groups than this paper. The contribution of the paper over existing work remains uncertain.


-  **Components and analyses already done in previous work.** One of the core components of the method is the *group attentioning* operation the authors introduce. However, this has been proposed previously over 4 years ago in [2, 3]. See, for example, how Fig. 3 in [2] summarizes Eqs. (1-8).

   Note in addition that [3] explicitly mentions the problem with squared images (Appx. C).

- **repeated analyses and conclusions.** I consider many of the statements included / derived in this paper to be well-known from literature. For example:
Line 316 - … we evaluate the linear class. acc. on both the NR and R datasets, which included images rotated in four directions. As shown GIE achieves highest accuracy on the R dataset. (Similar in lines 360 and 367). Results in Figure 4.
-> Please note that this is a direct effect of using G-CNNs. This has been known and proved for many years. For example, note how Fig 4 clearly resembles analyses done over 5 years ago by [4] (see Fig 4 in [4] ). It is unclear what the new insights are.
Analyses similar to those in Fig 5 were done in [1] as well. In fact, similar analyses have been done previously based on class-dependent statistics (not only dataset level statistics) in [5].

An additional concern relates to how the method is not comparing its results with previous related works, e.g., [1].

**Questions:**

### Questions and additional comments

- At the end of Sec 4.4, the authors have a paragraph depicting an extension to p8. But this paragraph does not summarize the findings of the result. It only states that it has been done.

- Why did the authors only restrained themselves to networks equivariant to discrete groups instead of continuous? Note that, it is even possible to define continuous equivariant networks that work with regular representations, e.g., [6].


### Conclusion

Due to the points outlined before, it is very difficult to gauge what the contribution and significance of this work is. I encourage the authors to spend time doing a thorough literature review, and outlining the differences of their method with existing approaches. Also, comparisons to more relevant methods should be considered in the experimental section.

Based on the previous analysis, I am unable to recommend acceptance at this point.

### References

[1] Winter, R., Bertolini, M., Le, T., Noé, F. and Clevert, D.A., 2022. Unsupervised learning of group invariant and equivariant representations. Advances in Neural Information Processing Systems, 35, pp.31942-31956.

[2] Romero, D.W. and Hoogendoorn, M., 2019. Co-attentive equivariant neural networks: Focusing equivariance on transformations co-occurring in data. arXiv preprint arXiv:1911.07849.

[3] Romero, D., Bekkers, E., Tomczak, J. and Hoogendoorn, M., 2020, November. Attentive group equivariant convolutional networks. In International Conference on Machine Learning (pp. 8188-8199). PMLR.


[4] Weiler, M., Hamprecht, F.A. and Storath, M., 2018. Learning steerable filters for rotation equivariant cnns. In Proceedings of the IEEE Conference on Computer Vision and Pattern Recognition (pp. 849-858).

[5] Urbano, A. and Romero, D.W., 2023. Self-Supervised Detection of Perfect and Partial Input-Dependent Symmetries. arXiv preprint arXiv:2312.12223.

[6] Finzi, M., Stanton, S., Izmailov, P. and Wilson, A.G., 2020, November. Generalizing convolutional neural networks for equivariance to lie groups on arbitrary continuous data. In International Conference on Machine Learning (pp. 3165-3176). PMLR.

---

### Note · Authors · 2024-11-27

**Comment:**

We would like to extend our sincere gratitude to all reviewers who took the time to read and provide feedback on our paper. After careful consideration, we have decided to withdraw our submission from ICLR. We recognize the need to further strengthen the experimental aspects of our work and clarify its distinctions from prior research. Thank you again for your valuable feedback, which will help guide us as we refine our work.

**Withdrawal Confirmation:**

I have read and agree with the venue's withdrawal policy on behalf of myself and my co-authors.